# Blockchain-Based Frameworks for Food Traceability: A Systematic Review

**DOI:** 10.3390/foods12163026

**Published:** 2023-08-11

**Authors:** Rizwan Matloob Ellahi, Lincoln C. Wood, Alaa El-Din Ahmed Bekhit

**Affiliations:** 1Department of Food Science, University of Otago, Dunedin 9054, New Zealand; rizwan.matloob@postgrad.otago.ac.nz; 2Department of Management, University of Otago, Dunedin 9054, New Zealand; lincoln.wood@otago.ac.nz; 3School of Management, Curtin University, Perth 6054, Australia

**Keywords:** food supply chain, Industry 4.0, food traceability 4.0, Web 3.0, blockchain, artificial intelligence, big data analytics, IFPS, metaverse, RFID, IoTs

## Abstract

With the rise of globalization and technological competition, the food supply chain has grown more complex due to the multiple players and factors involved in the chain. Traditional systems fail to offer effective and reliable traceability solutions considering the increasing requirement for accountability and transparency in the food supply chain. Blockchain technology has been claimed to offer the food industry a transformative future. The inherent features of blockchain, including immutability and transparency, create a dependable and secure system for tracking food products across the whole supply chain, ensuring total control over their traceability from the origin to the final consumer. This research offers a comprehensive overview of multiple models to understand how the integration of blockchain and other digital technologies has transformed the food supply chain. This comprehensive systematic review of blockchain-based food-supply-chain frameworks aimed to uncover the capability of blockchain technology to revolutionize the industry and examined the current landscape of blockchain-based food traceability solutions to identify areas for improvement. Furthermore, the research investigates recent advancements and investigates how blockchain aligns with other emerging technologies of Industry 4.0 and Web 3.0. Blockchain technology plays an important role in improving food traceability and supply-chain operations. Potential synergies between blockchain and other emerging technologies of Industry 4.0 and Web 3.0 are digitizing food supply chains, which results in better management, automation, efficiencies, sustainability, verifiability, auditability, accountability, traceability, transparency, tracking, monitoring, response times and provenance across food supply chains.

## 1. Introduction

Production of consistent quality and safe food that considers “farm-to-fork” production is the cornerstone of a successful food industry. Therefore, coordination of interdependent operations from raw-material manufacturing through to the delivery of the final product is a key component of supply-chain management [1]. The ever-evolving development of global food supply chains (FSCs) and marketplaces led to a massive increase in the trade of goods and information across international boundaries [2]. However, fraud, inefficient transactions, and suboptimal performance within FSCs have raised concerns regarding the authenticity and quality of goods, leading to an urgent need for better information exchange and credibility [3].

Food-supply-chain networks are impacted by diverse factors, such as regulatory policies, cultural norms, human behavior, and globalization. The complexity of these factors poses a significant challenge in analyzing information effectively and managing risks within the sector [4]. There is an urgent need to develop knowledge and technologies to address these challenges and ensure efficient supply-chain management. The international food supply chain has been experiencing tremendous pressure to improve transparency, facilitate trusted information exchange, and enhance the traceability of food products throughout the entire supply chain [5,6].

Businesses around the world have undoubtedly undergone a significant evolution, particularly in the realm of supply-chain management, since, traditional food supply chains have faced several challenges related to food safety, traceability, quality, fraud, inadequate monitoring, and insufficient policies, rendering them obsolete in today’s modern age [7]. Furthermore, the ever-increasing demand from consumers for year-round availability of food products has put immense responsibility on businesses to provide comprehensive information about product-specific attributes, including standards, safety, originality, accuracy, traceability, and provenance throughout the food supply chain [8].

The opacity of food-traceability systems raises concerns over proprietary and intellectual property, largely due to inadequate technology adoption and reliance on paper-based processes [9]. This can risk data accuracy and credibility when sharing sensitive information within the food supply chain. Consistent shared information is vital for successful product tracing and tracking, which traditional supervision systems like bar codes struggle with due to issues like data fragmentation, lack of accuracy, and interoperability [10]. Experts consider blockchain a solution for these FSC challenges, as it can transform supply chain design, organization, and lead to improved visibility and traceability [9].

Blockchain, a digital and decentralized ledger, allows secure information storage and sharing [11,12]. This immutable and transparent ‘chain of blocks’ is attractive for FSCs due to its traceability and resistance to tampering [5,6,7,8,9,10,11,12]. This modern innovation, combining different technologies, is garnering interest from academia and industry for its unique features like self-governance, anonymity, and security [13].

Introduced in 2009 by Satoshi Nakamoto, blockchain technology functions as a decentralized ledger, enabling consensus on data ownership within the network [14]. Nodes, or individual devices, within this network coordinate independently, eliminating the need for a centralized authority [13]. This technology provides a globally distributed database, controlled and shared by a collective, and is founded on a protocol resistant to human interference [15]. Food supply chains are actively exploring the potential of adopting blockchain technology to enhance their supply-chain management practices [16,17,18,19]. For example, Walmart’s successful blockchain project for monitoring the supply of pork in China and IBM’s blockchain-based food-tracking technology have been launched in 2016. Due to increasing momentum in blockchain adoption in the FSC, Albertsons, one of the most significant food retailers in the United States has joined IBM’s blockchain-based Food Trust network in 2019 together with other significant retail giants such as Walmart and Carrefour [19].

The intricate nature of FSCs suggests that the integration of this distributed ledger technology into the food supply chain can provide more transparency and accessibility of information. However, it shall be noted that there is a significant growth in digital technologies that are working with blockchain to create a synergistic effect and, thus, needs to be discussed.

Food businesses are exploring technologies like blockchain to enhance food safety and traceability. This is evident in the partnership between Walmart, IBM, and Tsinghua University that utilized blockchain for food safety in China [18], and the collaboration between Chinese E-commerce giant, Jindong, and local beef farmers to create a blockchain-enabled product database. Moreover, Alibaba has launched an initiative using blockchain to combat counterfeit food sales, partnering with international producers such as Australia’s Blackmores [20]. However, despite these efforts, blockchain implementations in food supply chains (FSCs) have faced criticism due to the absence of a comprehensive roadmap [21].

The importance of this study lies in exploring the growing use of blockchain in the food supply chain (FSC). Additionally, alongside blockchain, there are other emerging technologies within Industry 4.0 and Web 3.0 that could aid the FSC. These include artificial intelligence, big data analytics, RFIDs, NFC, IoTs, edge computing, cloud computing, among others, contributing to the technological growth supporting the FSC.

Although Gartner [22] forecasted the Top 10 Strategic Technology Trends for 2023, consisting of many upcoming Industry 4.0 and Web 3.0 technologies, most recent blockchain-based food traceability and supply-chain frameworks only incorporated a few technologies interoperable with blockchain [23,24,25,26,27,28,29]. However, a review of this body of literature shows the implementation of many of these emerging technologies, such as artificial intelligence, data analytics, edge computing, Digital Twins NTFs and metaverse, are either scant or absent in the current blockchain-based food-supply-chain frameworks.

We believe our research will make a significant impact by allowing researchers to explore the interoperability of blockchain with other emerging Industry 4.0 and Web 3.0 technologies. This will, in turn, equip them to develop more sophisticated frameworks for blockchain-based food supply chains.

The primary objective of this review is to thoroughly examine available literature on the implementation of blockchain technology in the food supply chain with the aim of identifying potential opportunities and novel pathways to implement this innovative technology in the FSC. Specifically, this review aims to examine currently available blockchain-based food-supply-chain frameworks and determine potential gaps in current blockchain-based food-supply-chain frameworks. This should enable identifying opportunities that are available for improved blockchain-based frameworks.

The subsequent sections of this research paper are as follows: a discussion about emerging technologies with respect to food supply chain and traceability followed by methodology, findings and discussion, limitations, conclusions, and Appendix A.

## 2. Emerging Technologies with Respect to Food Supply Chain and Traceability

Before assessing FSC models and frameworks, it is critical to understand the concepts of Industry 4.0 and Web 3.0. This understanding becomes particularly relevant as our systematic review of blockchain-based food traceability and supply-chain frameworks has revealed the integration of various Industry 4.0 and Web 3.0 technologies along with blockchain. These findings will be detailed further in the results and discussion section.

The first three industrial revolutions were marked using steam engines, electronics, and information technology, respectively. These revolutions led to the mechanization, mass manufacturing, and automation of production [30]. Industry 4.0, or IR4.0, signifies a new era of manufacturing characterized by the convergence of physical and digital worlds, termed as Cyber–Physical Systems (CPS). These systems enable interaction between digital infrastructure and physical objects, making information and services concurrently accessible for various purposes [31].

Industry 4.0 technologies encompass a range of innovations, including but not limited to additive manufacturing (3D Printing), artificial intelligence (AI), Global Positioning System (GPS), big data and analytics (BDA), Radio Frequency Identification (RFID), blockchain (BC), cloud computing (CC), Internet of Things (IoT), simulation, augmented reality, robotics, unmanned vehicles, and drones [32].This convergence is facilitated by numerous technologies, including a collection of internet-based technologies such as artificial intelligence (AI), machine learning (ML), big data, decentralized ledger technology (DLT), metaverse, decentralized applications (dApps), Semantic Web, edge computing, non-fungible tokens (NFTs), among others. These technologies are collectively referred to as Web 3.0. It’s important to recognize that Web 3.0 is still a developing concept, and its full realization has not yet been achieved. Various initiatives and projects are actively building the necessary infrastructure and tools for the decentralized web [33].

There are several websites and software programs that can intelligently process information, mimicking human intellect, which exemplify this concept. Web 1.0 was characterized by static websites and simple HTML pages, whereas Web 2.0 introduced interactive and dynamic content, social media platforms, and user-generated content. Web 3.0 aims to enhance the security and privacy of the decentralized Internet through the use of blockchain technology. In doing so, Web 3.0 seeks to address several of Web 2.0′s limitations and concerns, particularly in terms of data ownership, privacy, and centralized control [33].

Web 3.0, the next phase of internet evolution, is built on decentralized technologies like peer-to-peer networking and blockchain. This contrasts with the current Web 2.0, which is regulated by a few large tech companies, and offers a more user-centric, secure, and transparent experience [34]. Web 3.0 applications, termed as decentralized apps (DApps), function on decentralized networks made up of peer-to-peer nodes or blockchains. By eliminating centralized decision makers and regional restrictions, Web 3.0 has the potential to democratize financial systems [34,35].

AI and machine learning enable these systems to analyze data like humans, delivering personalized information without interference from major tech companies or compromising user data access and privacy. This shift to Web 3.0 could transform how people interact with the internet, fostering innovation, collaboration, and financial empowerment.

Key technologies enabling Industry 4.0 and Web 3.0 include the following:Near-field communication (NFC) enables quick, secure two-way transactions between electronic devices over short distances [36].Big data analytics extracts insights from complex data sets using advanced methods such as machine learning and data mining [37,38].Artificial intelligence (AI) mimics human intelligence through machines, computer systems, robotics, and digital devices [39].The Global Positioning System (GPS) provides accurate positioning and timing data, facilitating numerous applications like vehicle navigation and tracking systems [40,41].Cloud computing offers various remote computer services over the internet, such as virtual infrastructure and storage [42].QR codes are widely used for transmitting information in diverse areas, with varying error-correction levels [43,44].Radio Frequency Identification (RFID) is a flow-control technology that aids in the traceability of goods throughout the production chain [45].The Internet of Things (IoT) uses embedded devices to facilitate real-time communication and data exchange between smart devices [46].Edge computing executes computations near data sources, reducing response time and enhancing energy efficiency [47].Interplanetary File System (IPFS) supports the development of decentralized, globally addressable data structures [48].The metaverse is a network of interconnected virtual realms that coexist with the physical world, enhancing user interaction [49].

## 3. Methodology

To address the research questions, a systematic review approach was undertaken. A systematic review is a rigorous and comprehensive approach to analyze literature, allowing the identification of trends and patterns of information. We followed the guidelines outlined by Kitchenham and Charters [50] and Tranfield et al. [51], and adhered to the recommendations of Moher et al. [52] regarding the “Preferred Reporting Items for Systematic Reviews (PRISMA)”. Systematic reviews involve the use of explicit and reproducible methods to address predefined research questions by identifying, critically evaluating, and combining findings from primary research studies. As recently highlighted by Pollock and Berge [53], the key stages in conducting systematic reviews include defining the objectives and methods in a protocol, locating relevant research, gathering data, assessing the quality of the studies, synthesizing the evidence, and interpreting the results.

This systematic review adheres to a comprehensive approach to fulfill its objectives (Figure 1). To begin with, it formulates a well-defined research question that serves as a guiding principle throughout the review process. Subsequently, criteria are established to determine which studies should be included or excluded, ensuring that only pertinent ones are considered. The selection of literature involves a meticulous process of carefully choosing relevant studies and assessing their quality to guarantee reliability and validity. Once the literature is gathered, it undergoes thorough analysis and synthesis to uncover patterns, trends, and significant findings. Ultimately, the findings are disseminated through diverse channels, such as research articles, reports, or presentations, to effectively reach the target audience.

**Figure 1 foods-12-03026-f001:**
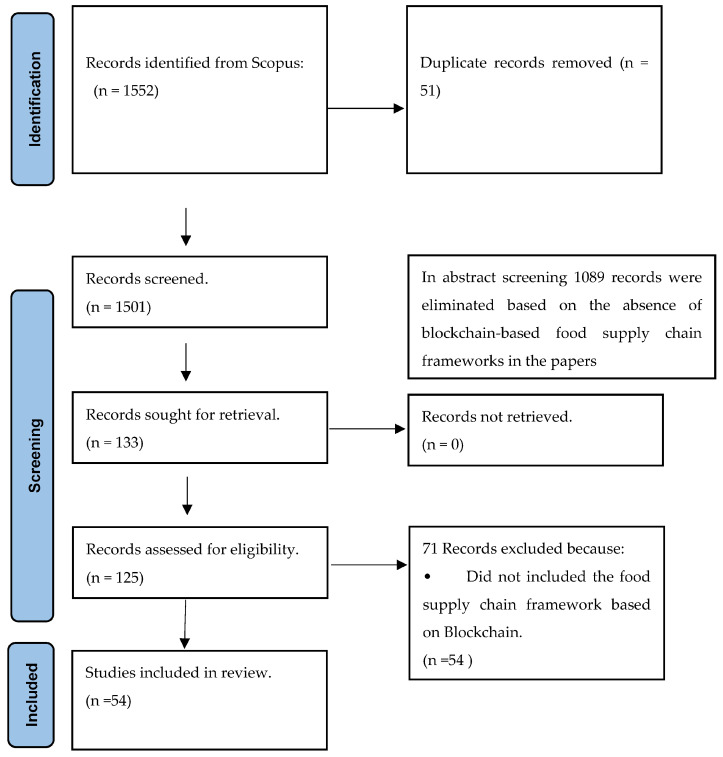
Preferred Reporting Items for Systematic Reviews and Meta-Analyses (PRISMA [54]).

This systematic review process guarantees a rigorous and methodical approach to acquire, evaluate, and present the existing evidence related to the application of interoperable technologies with blockchain for food traceability and supply chain.

### 3.1. Search Strategy

Scopus (the largest database for peer-reviewed research publications) was selected to obtain relevant published reports. The search string on Scopus was configured as Title-Abstract-Keywords:(TITLE-ABS-KEY (“Blockchain” OR “Block chain” OR “distributed ledger”) AND TITLE-ABS-KEY (“Food” OR “Meat” OR “Agri” OR “fruit” OR “Vegetable” OR “Grain” OR “Fish” OR “Honey” OR “Milk” OR “Dairy” OR “Tea” OR “Coffee”))

### 3.2. Inclusion Criteria

Inclusion and exclusion criteria were established to improve the focus of the review. The term “Blockchain” was introduced in 2008 but literature on the blockchain and the food supply chain was available only in 2016. However, as shown later in the descriptive statistics section of the paper, the literature on blockchain-based food supply chains has increased substantially since then.

All the publications from 2016 have been considered, which resulted in a total number of publications of 1552 in the selected string until 8 July 2023.

These documents were screened and only documents that provided a blockchain-based supply-chain framework for a food product or food in general were selected as eligible to be included in the research. Studies from languages other than English have been excluded.

## 4. Findings and Discussion

While interest in blockchain-based food systems started in 2016, it has since seen substantial growth, following an exponential curve. As per the search query, from a mere three publications in 2016, the number of publications on this topic surged to 422 per year by 2022. Moreover, a detailed analysis conducted in the Scopus database indicated that India and China were the principal contributors to this field, with India publishing 277 research papers and China contributing 226. Notably, despite Indian researchers producing the most publications, the most cited researchers on this subject were from China (Figure 2).

In 2015, The Economist dubbed blockchain technology as “The Trust Machine,” highlighting its capability to guarantee transparency and trust in data transactions. That same year, China grappled with a food safety issue involving the seizure of illegal livestock and products going back to the 1970s, unveiling complications within the food supply chain. To address these issues, Walmart employed blockchain technology to develop innovative solutions that ensured secure and transparent tracking of food items throughout the supply chain. In 2016, Walmart joined forces with IBM to create a more effective and efficient food traceability system based on Hyperledger Fabric. This system allowed Walmart to track mangoes in its US stores in just 2.2 s, a considerable improvement from the previous six-plus days, leading to enhanced transparency and trust in the supply chain.

The same year saw Walmart China launching the Walmart China Blockchain Traceability Platform, which successfully initiated and tested the first 23 product lines using VeChain’s blockchain technology. Moreover, Walmart conducted a trial of a blockchain-enabled end-to-end traceability system specifically for shrimp sourced from India and delivered to selected Sam’s Club stores in the United States [54].

Among the initial adopters of blockchain technology in the food sector were India, China, the United Kingdom, Italy, and the United States. At present, the United States is at the forefront of integrating blockchain technology in the food industry, with many companies using it to improve their supply-chain traceability systems. Growing concerns about food safety and the belief that blockchain can facilitate transparent supply chains have stimulated extensive research in this area. India, one of the world’s largest agricultural producers and a major contributor to its GDP, has also made significant contributions to the literature on blockchain-based food supply chains, along with researchers from China and Italy [55].

In 2016, the Chinese government reported 500,000 food safety inspection violations over a nine-month period. Additionally, during the recent China Food and Drinks Fair held in Tianjin, counterfeit products were rampant, leading industry experts to liken it to a “Disneyland of fakes”. China has a lengthy history of incidents involving food poisoning, malnutrition, and hospitalizations [56]. This includes various incidents, such as the presence of contaminated wheat protein, the use of sewage in tofu production, honey dilution with beetroot or rice syrup, toxic food incidents, the use of industrial salt, and baby formula contamination, among others. The need for improved food traceability in China arises from the urgency to rebuild trust after a long history of food fraud and safety issues [57]. In Italy’s context, a host of challenges in the agricultural supply chain, such as inadequate transparency, poor traceability, fragmentation, financial inefficiencies, and the ongoing risk of food fraud and safety concerns, have sparked considerable interest in exploring the potential benefits of using blockchain technology [58,59]

According to the selection criteria listed in the methodology section, the number of papers containing blockchain-based frameworks pertaining to food traceability and supply chains that have been included are displayed in Table 1. The table not only reports the frameworks available in the literature but also depicts the technologies that are used in these frameworks in conjunction with the blockchain.

Table 1 illustrates the application of Industry 4.0 and Web 3.0 technologies within blockchain-based models for food traceability and supply chains. It is clear that numerous technologies, such as Near Field Communication (NFC), big data analytics (BDA), and artificial intelligence (AI), have not been widely integrated into many of these frameworks. Given the growing adoption of Industry 4.0 and Web 3.0 technologies, as well as the emergence of new ones, it is highly likely that some of these technologies are missing from the current blockchain-based models for food traceability and supply chains. This points to potential areas for enhancement in these blockchain-based food traceability and supply chain models.

The inclusion of multiple Industry 4.0 and Web 3.0 technologies in food-traceability and supply-chain systems will be further examined below. This will help gain a better understanding of how blockchain-based food-traceability and supply-chain models can incorporate these technologies in an interoperable manner.

Interestingly, from the 54 frameworks selected for systematic review, only one food-traceability and supply-chain model, reported by Dey et al. [43], introduced the concept of a Web 3.0-enabled smart city that could use advanced technologies like blockchain, edge computing, artificial intelligence, cloud computing, and big data analytics to reduce food waste at different stages in the supply chain. This observation underscores a gap in the application of the term Web 3.0 among researchers studying food traceability and supply chains, suggesting its limited adoption in their research.

To provide a better outlook to the readers, the below table gives insights of the main applications along with the advantages and disadvantages of major Industry 4.0 and Web 3.0 technologies (Table 2 and Table 3). Please bear in mind that the level of security, scalability, and other parameters can vary depending on the specific implementations and configurations of these technologies.

Table 3 depicts the characteristics, evaluation and comparison table for major Industry 4.0 and Web 3.0.

Table 4 reflects the top five most comprehensive blockchain-based food traceability and supply-chain frameworks out of the total 54 identified through this literature review. Most of these frameworks, which incorporate maximum interoperable technologies, were published in 2022 and 2023. This indicates the increasing integration of Industry 4.0 and Web 3.0 technologies with blockchain, highlighting the growth in interoperability.

**Table 4 foods-12-03026-t004:** Blockchain-based food-traceability and supply-chain frameworks catering most of the interoperable Industry 4.0 and Web 3.0.

Title of Article/Report/Book	Author	Journal	Year	BC	RFID	IoT	QR	CC	GPS	NFC	BDA	AI
Blockchain-assisted internet of things framework in smart livestock farming	Mohammed Alshehri [60]	Internet of Things (The Netherlands)	2023	X	X	X		X	X		X	X
A blockchain-enabled security framework for smart agriculture	Chatterjee et al. [61]	Computers and Electrical Engineering	2023	X	X	X	X	X			X	
Construction of rice supply chain supervision model driven by blockchain smart contract	Peng et al. [62]	Scientific Reports	2022	X	X	X		X	X	X		
Blockchain-Enabled Supply Chain platform for Indian Dairy Industry: Safety and Traceability	Khanna et al. [63]	Foods	2022	X		X	X	X			X	X
Agriculture-Food Supply-chain management Based on Blockchain and IoT: A Narrative on Enterprise Blockchain Interoperability	Bhat et al. [64]	Agriculture (Switzerland)	2022	X	X	X	X	X	X		X	X
Applying blockchain technology to improve agri-food traceability: A review of development methods, benefits and challenges	Feng et al. [65]	Journal of Cleaner Production	2020	X	X	X	X	X	X			

Feng et al. [65] developed a blockchain-based traceability system for agriculture and food industries consisting of four layers:The business layer handles all supply chain activities, from farming to consumption, with each enterprise managing its traceability information.The IoT layer collects and records traceability data, such as quality and logistics, using devices like RFID and sensors that communicate with blockchain ledgers.The blockchain layer enhances transparency and security of food traceability using smart contracts, which enable real-time quality control and streamline planning processes.The application layer acts as an intermediary, facilitating access to detailed records of the logistics process, information, and capital flow [65].

Similarly, Chatterjee et al. [61] built a framework connecting stakeholders via blockchain technology in supply-chain networks. The process starts with the selection of “validator” nodes from regulatory agencies, tasked with authorizing transactions following quality standards verification. IoT devices facilitate automatic data collection and uploading to the blockchain network. Smart contracts ensure tamper-proofing and auto-resolution of contract violations, with blockchain networks operating self-sufficiently to authenticate product origin and purity [61].

Peng et al. [62] presented a dynamic supervision model for the rice supply chain, including an initialization module, a supervision module, and a storage module (Figure 3). The initialization module classifies participants into enterprises, consumers, and regulatory agencies, further splitting enterprises into six categories and sub-nodes. The supervision module uses smart contracts for real-time supervision of rice data. Data are stored in a blockchain cloud database and gathered via IoT devices such as RFID, NFC, mobile phones, computers, and GPS (Figure 4). This model ensures data integrity and dynamic supervision of collected data through blockchain and smart contract technology.

Bhat et al. [64] proposed the Agri-SCM-BIoT architecture, an innovative approach combining various Industry 4.0 and Web 3.0 technologies like RFID, QR codes, cloud computing, GPS, data analytics, and AI (Figure 5). This design addresses multiple challenges in the agriculture supply chain, such as storage and scalability optimization, interoperability, security, and privacy, enhancing transparency and traceability.

Similarly, Alshehri [60] developed a smart livestock farming model, integrating IoT and blockchain technologies to aid farmers. The model enables the creation of protective zones for livestock, notifies when boundaries are breached, and leverages sensors for real-time environmental data. Blockchain technology ensures product origin tracing, important for customer satisfaction and issue resolution [60].

### 4.1. Interoperable Web 3.0 and Industry 4.0 Technologies Used in Blockchain-Based Food Traceability and Supply-Chain Frameworks

Figure 6 shows a summary of technologies employed for interoperability within food traceability and supply-chain frameworks based on blockchain. Nevertheless, the question mark depicted in this figure draws attention to the web technologies that have yet to be explored in terms of their utilization in blockchain-based food traceability and supply chain system.

#### 4.1.1. Near-Field Communication in Blockchain-Based Food-Traceability and Supply-Chain Frameworks

Of the 54 frameworks (Table A1), only six incorporated Near-Field Communication (NFC) with blockchain for food supply chain and traceability. There has been growing emphasis on research into NFC-based energy harvesting in recent decades. The magnetic field generated by NFC is utilized not just for data transmission but also to power an integrated circuit containing a sensor module. This module can detect a range of environmental parameters, including temperature, soil, moisture, and pH. The NFC magnetic field is particularly used for data exchange. Energy harvesting based on NFC allows any NFC-enabled smartphone to be used as a data reader, thus eliminating the need for dedicated readers, and increasing efficiency [63,64,65,66].

Combining NFC and blockchain provides an opportunity to establish a more secure and efficient food supply chain. For instance, using NFC, a farmer can assign a unique identifier to each piece of produce, serving as a tracking mechanism that enables monitoring of the produce’s journey from the farm to the grocery store. By applying blockchain, pertinent data about the product such as its origin, cultivation location, and handling procedures can be recorded. This information becomes available to all participants in the supply chain, ensuring enhanced food-safety measures and traceability [67]. The combination of NFC and blockchain technologies fosters a more secure, efficient food supply chain, enhancing food safety, traceability, and transparency.

#### 4.1.2. Big Data Analytics in Blockchain-Based Food Traceability and Supply-Chain Frameworks

Only 12 studies have thus far incorporated big data analytics in their blockchain-based food-traceability and supply-chain systems. The melding of these technologies offers a significant opportunity to reshape the food supply chain [37,38]. Notably, data analytics, which can gather large volumes of data from sensors, RFID tags, and GPS devices, is anticipated to revolutionize animal farming by fostering accurate and efficient practices. This approach facilitates comprehensive monitoring of food product flow, identifying wastage hotspots, and enhancing food safety measures [63,64,65,66,67,68].

Additionally, leveraging historical data allows for the efficient monitoring of large cattle herds and simpler identification of health issues based on previous symptom records [60]. Further innovation comes from wearable IoT devices that monitor cattle behavior and health conditions, helping detect diseases early and taking necessary actions for animal well-being.

Collecting environmental data through IoT devices also provides insights for optimizing farming practices, including irrigation and fertilization decisions, thereby improving farm productivity [60,61,62,63,64]. The marriage of blockchain and big data secures data exchange and provides timely demand information, drawing significant attention from stakeholders and promising substantial benefits in risk minimization [69].

The synergy of big data analytics and blockchain creates a precise understanding of the food supply chain, serving as a basis for informed decisions in food production, distribution, and marketing [37,38]. Massive food-safety data can be efficiently processed, assisting in microbial risk assessment, outbreak prevention, and pathogen trend identification, thus improving food-safety outcomes [70,71].

The future of food safety is significantly influenced by big data analytics. Benefits include real-time food monitoring during storage and transport, synchronized digital labeling integrated with cloud information, and enhanced traceability with blockchain, all contributing to a more robust food-safety framework [38].

Practical applications are already seen in China. Zhongnan Group and HeiLongJiang Agriculture Company Limited collaboratively enhanced agri-food traceability using big data and blockchain [72]. Likewise, JD.com, along with various food companies, launched a blockchain-based traceability platform, ensuring food quality traceability from farm to table. Other notable service providers include IBM, CyberSecurity, Arcnet, and ripe.io, serving companies like Walmart, Nestle, Unilever, and Tyson Food. As both big data and blockchain are integral to China’s national strategy, their combined use in agricultural product tracing is now standard, highlighting their exceptional benefits [69].

#### 4.1.3. Artificial Intelligence in Blockchain-Based Food Traceability and Supply-Chain Frameworks

Artificial Intelligence (AI), a pivotal technology in the era of Web 3.0 and Industry 4.0, features in 13 of the 54 frameworks selected for food traceability and supply-chain management. Its critical role in contemporary agriculture spans various farming aspects, from fostering healthier crop cultivation, managing pests effectively, to monitoring soil conditions and improving overall food-supply-chain management. Integrating AI technologies reaps substantial rewards in refining agricultural processes, boosting crop quality, and optimizing resource use, driving advancements in the agricultural industry [73].

AI aids in determining optimal seed planting times by analyzing environmental factors and suggesting the most suitable seeds per specific conditions [74]. Additionally, AI-powered weather forecasts offer substantial benefits to agriculture. AI algorithms provide farmers with accurate, timely weather predictions, facilitating informed decision making and planning for farming activities. Thus, the incorporation of AI in agriculture offers tools to enhance productivity and efficiency while reducing weather-related risks [75].

Farmers can improve crop quality and reduce time-to-market through the application of artificial intelligence (AI). AI’s significant contribution lies in delivering insights into soil properties, guiding farmers on the optimal fertilizer for soil quality enhancement. AI-enabled analysis empowers farmers to make informed decisions, driving soil health and subsequently boosting crop yield and quality. The integration of AI in agriculture equips farmers with the tools and information for soil management optimization, leading to sustainable and efficient crop outputs [73].

Blockchain technology equips AI with reliable data, integral to its “deep learning” process. These advanced algorithms help enhance system scalability and optimize energy consumption. AI, supported by blockchain, can identify security vulnerabilities, thereby bolstering optimization [76]. As a robust platform, blockchain channels critical data to AI, refining decision-making accuracy and ensuring a secure, decentralized AI system. This collaboration heightens food traceability [76].

Machine learning (ML) is being embraced across the agricultural supply chain, from preproduction to distribution. During preproduction, ML assists in crop-yield forecasting, soil evaluation, and irrigation-needs assessment. It aids in disease detection and weather forecasting during production [77]. In processing stages, ML helps forecast production planning for quality products.

In distribution, ML finds use in areas like storage, transportation, and consumer analysis [78]. By amalgamating data such as equipment needs and nutrient info into blockchain-powered models, precision agriculture tools aid stakeholders in crop-yield-forecasting decisions, maximizing agricultural outcomes.

#### 4.1.4. GPS in Blockchain-Based Food-Traceability and Supply-Chain Frameworks

This systematic review highlights the integration of Global Positioning Systems (GPS) in blockchain-based food-traceability and supply-chain frameworks. Among the 54 frameworks, 14 combine GPS and blockchain for food-supply-chain traceability. GPS allows real-time tracking of food products, ensuring their secure transportation and storage. In case of accidents, products can be traced back via GPS coordinates at each supply-chain stage. The system can also alert relevant authorities about deviations from preset conditions in the cold chain distribution. By storing GPS data on the blockchain, monitoring food product movement from farm to consumer is feasible, helping identify issues like foodborne illnesses or contaminated items [40,41].

As products travel the supply chain, ownership shifts from processors to distributors, a process automated and documented via blockchain-based smart contracts. These contracts also log any abnormalities during distribution. Utilizing GPS, distributors input these data into their application. Combining blockchain and GPS optimizes supply-chain management, providing real-time product location visibility to stakeholders, which aids in optimizing logistics and minimizing losses [24].

Packages equipped with sensors, cameras, and GPS enable product condition monitoring during transport, with data stored immutably on a blockchain ledger for access by all involved parties. By integrating blockchain and GPS, supply-chain management is streamlined, with stakeholders gaining real-time product location insights to optimize logistics and minimize losses [41].

#### 4.1.5. Cloud Computing in Blockchain-Based Food Traceability and Supply-Chain Frameworks

Out of 54 blockchain-based food traceability and supply-chain frameworks, 23 incorporate cloud computing. Cloud computing offers a flexible platform for handling large agricultural data from various IoT sources, like sensors and drones. It allows farmers and organizations to store and access data securely, facilitating easier data analysis and decision-making [79]

Cloud computing enables the processing and analysis of agricultural data, offering insights from advanced analytics, machine learning, and AI algorithms. These insights assist farmers in decision-making for crop management and resource use, boosting production and efficiency [80]. IoT devices linked to the cloud allow real-time tracking of crop health and environmental conditions, enabling rapid, data-driven responses. Transactions on a blockchain are secured with digital signatures and unique hash codes. However, the storage needs of all interconnected blocks in a blockchain necessitate the consideration of both distributed systems and cloud computing during implementation [81].

Central cloud servers in a blockchain structure facilitate communication across the food production cycle and enable supply chain expansion as needed. They allow consumers to verify livestock conditions through a smartphone interface connected to the cloud server, validated by the blockchain [82]. The integration of RFID, sensors, blockchain, and cloud servers enhances overall food-traceability and supply-chain effectiveness. Users can access information stored in the cloud, such as product details, through websites or barcode scans [83]. Cloud computing and big data are transforming the food industry, providing a base for data collection and analysis throughout the supply chain, and offering quick messaging services [84]. IoT sensors monitor food freshness at each point of sale, sending data to the cloud and blockchain ledger for transparency [84]. Cloud-based technology, interconnected with blockchain and IoT, enables comprehensive monitoring and real-time inventory management. The integration of blockchain and IoT shows potential in terms of scalability, security, and efficiency [85].

#### 4.1.6. QR Codes in Blockchain-Based Food-Traceability and Supply-Chain Frameworks

Many frameworks integrate QR codes and blockchain for food traceability and supply chains, enabling easy access to product information anytime, anywhere (Table 1). Blockchain and QR codes digitize food supply chains, as demonstrated by studies and applications like TE-FOOD, adopted by Ho Chi Minh City for pork tracking. This transparent data system includes all supply-chain participants and utilizes a mobile app to assign unique IDs and QR codes at each process stage. Customers can verify the meat’s origin by scanning the label, allowing authorities to confirm its origin and quality [86]. Since its launch in 2017, TE-FOOD has expanded to track poultry and eggs, training over 6000 companies in South Vietnam [86]. Nestlé and Walmart also utilize QR codes and blockchain for the traceability of coffee beans and mangoes, respectively. This combination of technologies improves food tracking, enhancing supply-chain transparency [87].

#### 4.1.7. Radio-Frequency Identification (RFID) in Blockchain-Based Food-Traceability and Supply-Chain Frameworks

RFID technology, prominently used with blockchains, plays a pivotal role in enhancing food safety and traceability. Tian [88] proposed a model using these technologies to streamline data acquisition, dissemination, and sharing across various supply-chain links. This model ensures the authenticity and integrity of shared information. Similar to Tian’s work, Mondal et al. [89] presented a “Blockchain-Inspired RFID-Based Information Architecture” enabling real-time quality monitoring and unique product identification.

The role of RFID in livestock monitoring was explored by Yang et al. [82], demonstrating the feasibility of an integrated system for early detection of abnormal behavior, thus minimizing losses for producers. Another method was presented by Lin et al. [90]. which employs blockchain and the Ethereum platform for product traceability. The study highlighted the role of RFID in ensuring data immutability and fraud prevention in Taiwan’s food industry.

RFID and blockchain integration provide supply-chain-wide visibility of food products [91]. The Global Traceability Standard uses these technologies to allocate unique identifiers throughout the supply chain [17]. After harvesting, RFID-tagged commodities preserve crucial data, including environmental factors and cropping conditions [92].

Retailers acquire comprehensive supply chain information and consumers can access this through RFID readers. The traceability system allows real-time auditing and swift identification of faulty items [93]. Despite some security concerns, the combination of RFID and blockchain technologies provides a flexible, automated, and immutable food-traceability system [81].

#### 4.1.8. IoT in Blockchain-Based Food-Traceability and Supply-Chain Frameworks

The fusion of blockchain and Internet of Things (IoT), two noteworthy innovations, yields numerous advantages in the agricultural sector [94]. IoT applications enhance productivity and efficiency in agriculture by enabling optimal farming procedures, facilitating the identification of field variables, monitoring environmental conditions, and guiding crop selection and rotation [95].

IoT assists in various agricultural processes, including crop seeding, irrigation, harvesting, and transportation, by utilizing sensors for effective monitoring and control of operations [96]. The integration of blockchain with IoT is predicted to significantly contribute to the global economy, with contributions reaching $3 trillion by 2030 [97].

IoT sensors enable automated recording and real-time data acquisition on food products, improving their quality assessment [98]. When coupled with blockchain, IoT facilitates error elimination and enhances surveillance, transaction, and data-tracking efficiency [84].

Tsang et al. [99] introduced a BLC-IoT mechanism for reliable and efficient monitoring in the food supply chain (FSC), addressing privacy and security concerns [100,101]. In another study, Osmanoglu et al. [102] suggested remote management and secure information transactions in the FSC by integrating IoT with a distributed ledger system. The adoption of BLC and IoT technologies fosters an electronic culture and promotes smart agricultural practices, contributing to food security and customer service enhancement [103,104].

### 4.2. Interoperable Web 3.0 and Industry 4.0 Technologies Missing or Rarely Applied by Blockchain-Based Food-Traceability and Supply-Chain Frameworks

#### 4.2.1. Edge Computing in Blockchain-Based Food-Traceability and Supply-Chain Frameworks

Edge nodes are essential in providing the cloud’s blockchains with significant amounts of data, especially for services with high real-time or bandwidth demands. Both edge and cloud environments can accommodate artificial intelligence models and distributed agents. The cloud side optimizes supply-chain models for organic products, order management, and logistics, while the edge side uses predictive capabilities to estimate crop yield and quality based on light, water, fertilizer, and other growth adjustments [105].

Due to the rapid expansion of IoT, billions of smart devices are installed every year, which has led to a surge in sensor deployment and a massive increase in data volume. This growth in data strains cloud-server resources, resulting in delayed response times [106]. Edge computing models are therefore necessary to ease the load on cloud servers.

According to Hu et al. [105], combining blockchain with edge computing can revolutionize IoT in agriculture, necessitating a shift in the computational model to optimize local and cloud resources. Blockchain systems should be deployed in the cloud for extensive data mining, while data from peripheral devices should be preprocessed locally and transmitted to the cloud in a compressed form.

Augmented reality (AR) has numerous applications in agriculture, and it is predicted to be significant in agricultural IoT. Augmented reality systems assist in farming operations like pest identification, crop treatment monitoring, and herbicide selection [107,108]. Due to increasing user expectations for real-time efficacy and the cloud’s limitations to meet VR/AR computing’s higher bandwidth requirements, edge computing is preferable to cloud computing.

Augmented reality applications in agriculture, such as those presented by] Neto & Cardoso [109] and Liu et al. [69], use sensor networks for tasks like fungus detection and simulation of plant and livestock growth. Huuskonen and Oksanen [110] proposed a precision fertilizer application control using drones and augmented reality, demonstrating how AR and VR technologies combine virtual and real environments in agricultural applications [111].

Furthermore, Premsankar et al. [112] showed that cloud deployments have higher latency than edge computing, which led to the development of edge computing as a solution.

There is potential of edge computing in agriculture when combined with AI, blockchain, and VR/AR technologies. In the context of artificial intelligence, edge computing aids in data preprocessing and shared computation between peripheral devices and cloud servers. For blockchain technology, edge computing addresses the limitations in computation capacity and energy availability of terminal devices.

#### 4.2.2. Interplanetary File System in Blockchain-Based Food-Traceability and Supply-Chain Frameworks

Many blockchain projects heavily rely on the InterPlanetary File System (IPFS) for off-chain data storage, offering a solution to problems like centralized control and ambiguous data in traditional traceability systems [26]. Outshining other distributed file structures, IPFS provides a potent, open-source storage system that effectively manages user content across the network [113].

Blockchain technology, as used in a monitoring system developed by Babu and Devarajan [26], enhances traceability data’s transparency and reliability. The authors store encrypted data both in blockchain and IPFS (Figure 7) to streamline information retrieval and reduce blockchain load. This technology, coupled with recording agricultural data in IPFS, lays the groundwork for trustworthy food supply chains and data-driven agricultural techniques, while mitigating blockchain storage inflation [26].

Merging blockchain and IPFS in food supply chains increases transparency, traceability, and accountability, fostering trust among participants and enhancing consumer safety. Yang et al. [114] proposed using Hyperledger for database processing to overcome blockchain’s limited storage, despite its shortcomings such as high cost and slow data transfer. Liao and Xu [115] and Xie et al. [116] developed blockchain-based monitoring systems for agricultural quality control using sensor networks and IoT technology, respectively. However, storing extensive information on blockchain introduces network overhead.

Cocco et al. [117] presented a hybrid model, integrating blockchain and IPFS for quality control in food supply chains. Their Ethereum-based application stores only essential verification data on blockchain, with certificates stored and validated in IPFS.

Wang et al. [118] suggested a framework to enhance traceability, shareability, and data security in supply chains, using consortium models and smart contracts, reducing reliance on centralized institutions.

#### 4.2.3. Metaverse Computing in Blockchain-Based Food-Traceability and Supply-Chain Frameworks

The metaverse is characterized by social and cultural interactions that can be financial or political in context, mirroring a simulated version of reality. The diagram below visually represents the interconnectedness and convergence between the physical world and the metaverse (Figure 8). As an alternative world, the metaverse offers possibilities that cannot be effectively realized within the confines of the actual world [119].

During the COVID-19 pandemic, UC Berkeley students used Minecraft to host a virtual commencement ceremony, demonstrating the increasing convergence of offline and online worlds. Roblox further exemplifies this by allowing game development within its platform, with in-game tokens exchangeable for real currency [121].

The Fourth Industrial Revolution has expanded the virtual realm significantly, but it raises concerns about identity misrepresentation and misinformation. For a reliable metaverse, F establishing a trustworthy foundation is crucial, which is where blockchain technology steps in, fostering trust and security [122].

In the metaverse, users can exchange goods and services with bit-based currency, substituting physical currency and freeing activities from real-world limitations. However, data reliability is paramount [120].

Integrating food traceability in the metaverse connects the real and virtual worlds, extending the food experience beyond physical consumption. With blockchain’s secure record keeping, food-supply-chain data can be accurately logged, and smart contracts can ensure food safety compliance [123]. Covici et al. [124] suggest five design directions for a food-centered metaverse (Figure 9).

Blockchain technology is vital for the metaverse’s economy, preventing centralized control and supporting recognition of value for in-metaverse goods, essential for real-world-like transactions. The arrival of NFT-based blockchain tech and WEB 3.0 further energizes the metaverse, making its realization increasingly plausible. Individuals can create unique avatar characters or items, with their uniqueness validated by NFT technology [120].

We posit that the metaverse can bridge the digital and physical worlds, offering new opportunities for supply-chain stakeholders, including customers. Blockchain technology can contribute significantly to metaverse development by boosting transparency, trust, and efficiency.

#### 4.2.4. Other Missing Technologies in Blockchain-Based Food-Traceability and Supply-Chain Frameworks

Apart from previous technologies, other Industry 4.0 and Web 3.0 technologies like DeFi, DApps, NFTs, and Digital Twins could synergize with blockchain, but their use in the food industry is limited, resulting in sparse literature for review. However, given ongoing tech advancements and growing research on blockchain-based food traceability, this systematic review can guide future research in blockchain-based food traceability and supply-chain systems.

## 5. Limitations of Our Work

It is important to note that there may be emerging technologies focused on food traceability and transparency that incorporate blockchain but were not included in our review. This limitation stems from our search strategy, which specifically targeted literature related to blockchain-based food-supply-chain traceability frameworks. Additionally, as discussed earlier, the development and application of Industry 4.0 and Web 3.0 technologies, such as DeFi, DApps, NFTs, and Digital Twins, in the context of food supply chains are still in their early stages. Therefore, our research suggests that leveraging multiple technologies and addressing the identified gaps in the literature review would be beneficial in understanding the current state of development in food supply chains.

## 6. Conclusions and Future Work

The findings demonstrate the significant role of blockchain technology in improving food traceability and supply-chain operations, as evidenced by its application in various research studies. Furthermore, the paper explores the compatibility of blockchain with other Industry 4.0 and Web 3.0 technologies, providing valuable insights into the synergies between these technologies. It highlights how food supply chains are being transformed and modernized to facilitate enhanced information sharing while ensuring security, reliability, and transparency throughout the entire supply chain.

This paper serves as a valuable resource, providing new perspectives and directions for future scholars to enhance the transparency and traceability of food supply chains by utilizing interoperable technologies with blockchain. Currently, most of the blockchain-based food-traceability and supply-chain frameworks, including the recent one, fail to implement the interoperability of blockchain with emerging Industry 4.0 and Web 3.0 technologies. The reason for such a gap in the application of technology is because Web 3.0 and Industry 4.0 technologies are still in development and the application of these technologies is yet to be incorporated with respect to the food supply chain.

Our research provides a valuable contribution by opening new avenues of research to explore the interoperability of blockchain technology with emerging Industry 4.0 and Web 3.0 technologies and will enable researchers to delve into the interoperability of blockchain technology with emerging technologies. This, in turn, would empower future researchers to develop more advanced frameworks for blockchain-based food supply chains.

It is evident that the application of blockchain in food supply chains is still in its nascent phase and continuously evolving. With the integration of blockchain and other emerging Industry 4.0 and Web 3.0 technologies, we can expect remarkable advancements in revamping food supply chains.

## Figures and Tables

**Figure 2 foods-12-03026-f002:**
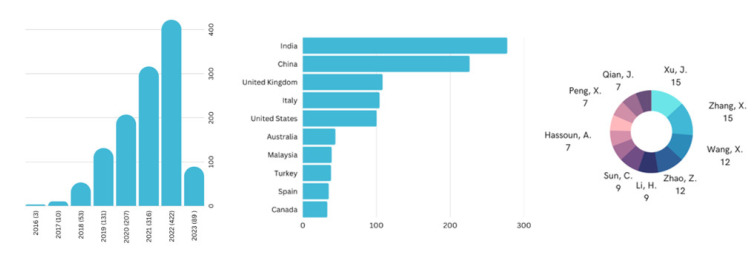
Growth of literature, top countries contributing to research and top researchers on blockchain-based food traceability and supply chain. The circle chart depicts the number of publications by leading authors in this field. Both J. Xu and X. Zhang are the top researchers on blockchain-based food traceability and supply chain.

**Figure 3 foods-12-03026-f003:**
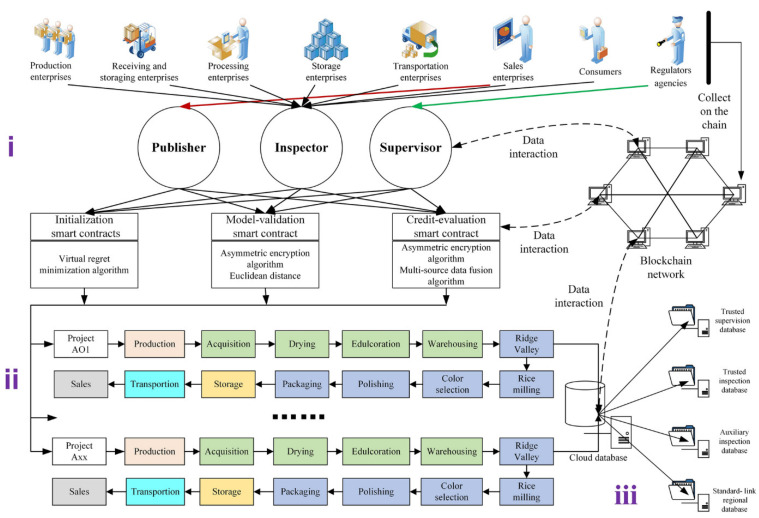
Dynamic supervision model for the rice supply chain [62]. Used under a Creative Commons Attribution 4.0 International License.

**Figure 4 foods-12-03026-f004:**
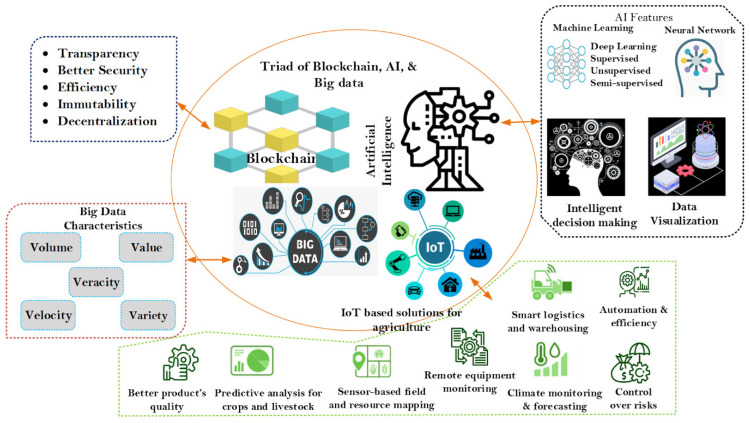
Characteristics of blockchain, Internet of Things, artificial intelligence, and big data analytics for smart farming [64]. Used under a Creative Commons Attribution 4.0 International License.

**Figure 5 foods-12-03026-f005:**
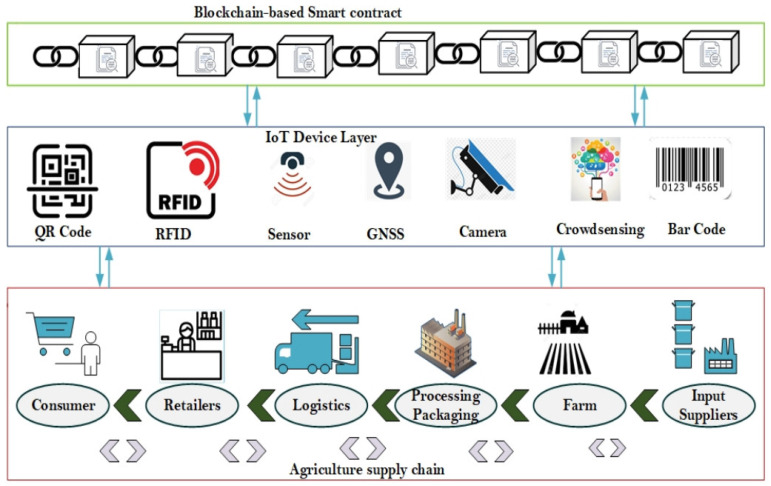
Smart agriculture supply chain [64]. Used under a Creative Commons Attribution 4.0 International License.

**Figure 6 foods-12-03026-f006:**
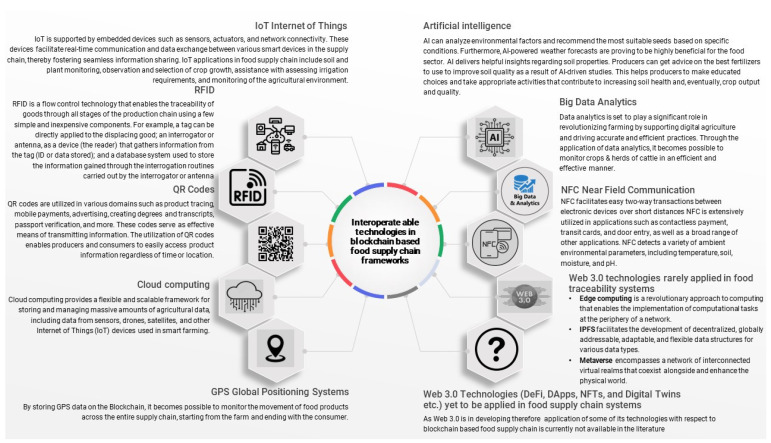
Interoperable Web 3.0 and Industry 4.0 Technologies used in blockchain-based food-traceability and supply-chain frameworks. (Source: Authors’ own work).

**Figure 7 foods-12-03026-f007:**
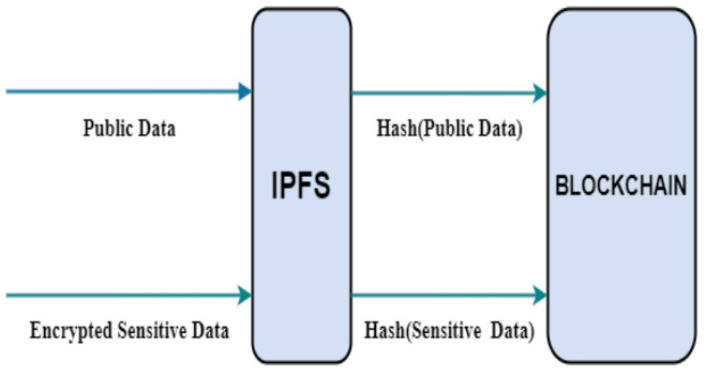
A hybrid system of Blockchain and IPFS system for information in agricultural food supply network [26]. Open-access license: https://thesai.org/Publications/IJACSA (accessed on 15 May 2023).

**Figure 8 foods-12-03026-f008:**
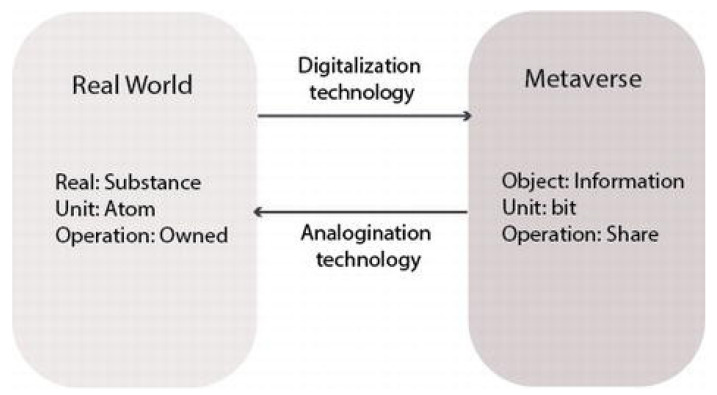
Connection between real world and metaverse [120]. Licensed under a Creative Commons Attribution 3.0 International License.

**Figure 9 foods-12-03026-f009:**
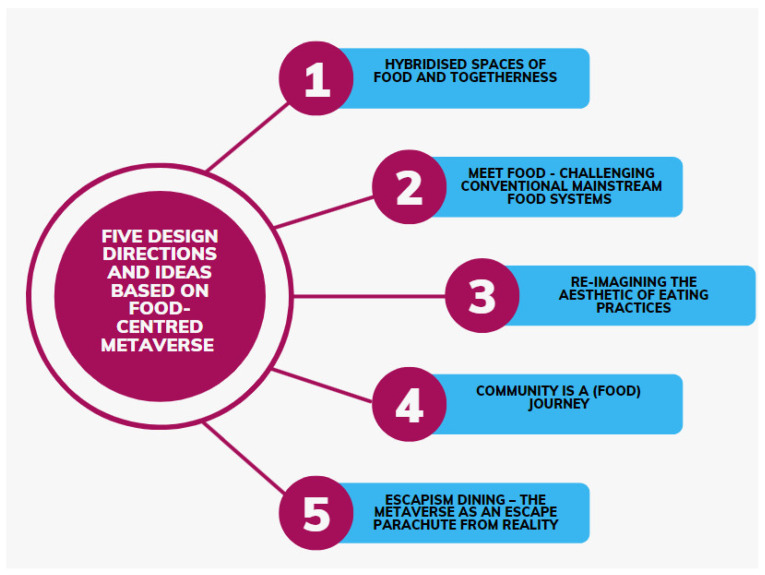
Five design directions and ideas based on food-centered metaverse [124]. (Source: Authors’ own work).

**Table 1 foods-12-03026-t001:** Interoperable Industry 4.0 and Web 3.0 Technologies are used by blockchain-based food traceability and supply-chain frameworks identified by this systematic review.

Industry 4.0 and Web 3.0 Technologies Used with Blockchain								
Blockchain and Smart Contracts	IoT	RFID	QR	CC	GPS	AI	BDA	NFC
54	37	26	23	23	14	13	12	6

**Table 2 foods-12-03026-t002:** Overview of applications, advantages and disadvantages of major Industry 4.0 and Web 3.0 technologies.

Technology	Description	Main Applications	Advantages	Disadvantages
**Blockchain**	A decentralized digital ledger for secure transactions and data storage.	Cryptocurrency, supply-chain management, smart contracts, finance	Transparency, security, immutability.	Scalability, energy consumption, storage requirements, regulatory concerns.
**IoT**	An interconnected network of physical devices and sensors that communicate and exchange data.	Smart homes, industrial automation, healthcare monitoring.	Automation, real-time insights, efficiency.	Security vulnerabilities, data privacy, interoperability.
**RFID**	Uses radio waves to identify and track objects with tags containing electronic information.	Inventory management, asset tracking, and access control.	Efficiency, real-time tracking, and reduced manual effort.	Cost, limited range, potential interference.
**QR Code**	Two-dimensional barcode that can store various types of data.	Marketing campaigns, ticketing, and payment systems.	Versatility, easy scanning, and high storage capacity.	Limited data capacity and scanning limitations in certain conditions.
**Cloud Computing**	Delivery of on-demand computing resources over the internet.	Infrastructure as a Service (IaaS), Platform as a Service (PaaS), and Software as a Service (SaaS).	Scalability, cost-effectiveness, accessibility.	Data security, vendor lock-in, and potential downtime.
**Artificial Intelligence**	Simulation of human intelligence in machines for autonomous learning and decision making.	Natural language processing, computer vision, recommendation systems.	Automation, improved efficiency, advanced analytics.	Ethical concerns, bias, job displacement.
**Big Data**	Large and complex data sets that require specialized processing techniques.	Business analytics, predictive modelling, personalized recommendations.	Insights generation, competitive advantage.	Data privacy, storage infrastructure, data quality.
**GPS**	Global navigation satellite system for location and timing information.	Navigation, logistics, and geolocation services.	Accuracy, real-time tracking, and widespread availability.	Signal limitations indoors or in remote areas.
**NFC**	Short-range wireless communication for contactless data exchange.	Mobile payments, access control, and ticketing systems.	Convenience, simplicity, and compatibility with smartphones.	Limited range, security concerns, adoption barriers.

**Table 3 foods-12-03026-t003:** Characteristics of major Industry 4.0 and Web 3.0 technologies.

Technology	Security Features	Connectivity	Scalability	Data Handling
**Blockchain**	Offers high-level security with decentralized networks	Enables connectivity among multiple devices	Scales well for transactions and data storage	Handles data immutability and transparency through a ledger
**IoT**	Exhibits varying levels of security	Facilitates connectivity across diverse devices and protocols	Scales effectively for large-scale deployments	Manages real-time data collection and analysis
**RFID**	Provides basic security measures	Utilizes short-range wireless communication	Supports scalability for tracking multiple items	Offers limited data storage capacity
**QR Code**	Lacks inherent security features	Enables connectivity through scanning	Not applicable for scalability	Encodes and retrieves data efficiently
**Cloud Computing**	Implements robust security measures	Utilizes internet-based connectivity	Highly scalable for resource allocation	Manages remote data storage and processing
**Artificial Intelligence**	Security implementation varies	Relies on internet-based connectivity	Scales effectively for data processing and analysis	Handles large-scale data processing and learning algorithms
**Big Data**	Security implementation varies	Relies on internet-based connectivity	Highly scalable for handling large datasets	Manages data storage, processing, and analysis
**GPS**	Lacks inherent security features	Utilizes global positioning and satellite connectivity	Not applicable for scalability	Tracks location-based data effectively

## Data Availability

The data presented in this study are available within the article.

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
