# Peer review of "Blockchain-Based Frameworks for Food Traceability: A Systematic Review"

_foods, 2023, doi:10.3390/foods12163026_

Round 1

Reviewer 1 Report

The paper focuses on a comprehensive review of the blockchain-based food supply chain frameworks to identify the available solutions and the scope for future improvements. 

The abstract needs improvement. It primarily focuses on research work. I would suggest indicating some of the challenges so that the readers get an understanding of the necessity of the research.

The Introduction needs improvement. It is too broad and needs to be concise. Unnecessary texts should be removed to focus on the core aspects. Also, rather than asking research questions, I would suggest defining the key contributions. Additionally, it would be beneficial to include a paragraph, at the end, that provides an overview of the content in the subsequent sections. This will guide the readers on what to expect and how the paper is structured.

Section 2 lacks substantial and insightful content.

Section 3 lacks clarity and precision, containing excessive and irrelevant context.

The authors seem more focused on expanding the paper rather than conducting a thorough analysis based on their investigation.

I would suggest reworking this section by providing a concise and focused analysis. Also, presenting the findings in a structured format would greatly enhance the readers' understanding of the research output.

Such a broad conclusion is not necessary. The conclusion should solely focus on the research output. Also, please use the appropriate tense in the conclusion. Furthermore, the future scope of the research is not very clear.

Author Response

Dear the Editor and Reviewer 1,

Thank you very much for your time and effort in reviewing our paper that improved our MS. We carefully addressed the comments and responded to all queries.

All new and amended text are highlighted in yellow for ease of tracking.

Shifted and reduced text is highlighted as blue in the manuscript.

Deleted text is stricken and is coloured red.

Best regards,

The Authors

Reviewer 1

The abstract needs improvement. It primarily focuses on research work. I would suggest indicating some of the challenges so that the readers get an understanding of the necessity of the research.

Response:

Noted with thanks. The abstract has been updated to depict the challenges in this research area and the following information has been added:

With the rise of globalization and technological competition, food supply chain has grown more complex due to the multiple players and factors involved in the chain. Traditional systems fail to offer effective and reliable traceability solutions considering the increasing requirement for accountability and transparency in the food supply chain.

The Introduction needs improvement. It is too broad and needs to be concise. Unnecessary texts should be removed to focus on the core aspects.

Response:

Thank you for your suggestion., In order to make the introduction concise the following information has been removed from the introduction:

The complexities of managing food supply chains are characterized by the need to integrate diverse tasks that are interdependent on each other (Rejeb & Rejeb, 2020, Longo et al., 2019). Traceability and transparency within supply chains have become important requirements, considering several scandals that have emerged in global food supply chains such as the horsemeat scandal in Europe and the melamine fraud in China which raised concerns and highlighted the need for robust traceability. As a result, regulations now mandate that every ingredient in a food product must be traceable to its source (Li et al., 2021; Barnard & Connor, 2017).

Manufacturers face the arduous task of delivering high-quality products at competitive prices, while simultaneously keeping operating expenses to a minimum and meeting customer expectation of secure and timely delivery of products. To achieve these goals, many food processing companies, especially multi-national ones, resort to outsourcing certain sections of their supply chain operations to third-party companies or strategically establishing their manufacturing and distribution hubs in low-cost regions and thus increasing the number of locations and intermediaries adding more complexity to the supply chain (Saha et al., 2022).

The emergence of UN sustainable development goals and One Health, which is advocating the integration of human, animal and environment wellness, concepts suggested that there are a lot of health and safety issues that must be resolved by creating sustainable food ecosystems. Blockchain can provide a platform to tackle some of these issues. For instance, the "Food Watch" program in Dubai digitalizes the roles and procedures involved in food safety using blockchain and other internet-based technology. Food Watch was launched in 2018 by Dubai Municipality's Director General, Hussain Nasser Lootah as a digital platform with the goal of fully digitizing the food safety and nutritional details of all food items offered in the Emirates' 20,000 or more food establishments. This service's goal is to provide users with the knowledge they need to choose the right diet for themselves (IBM, 2018; Detwiler, 2018).

Also, rather than asking research questions, I would suggest defining the key contributions.

Response:

We appreciate your recommendation to highlight the key contribution. Accordingly, we have added the key contribution of our research as follows:

“The significance of this research lies in examining the increasing adoption of block-chain technology in the FSC. Furthermore, there are other technologies emerging within Industry 4.0 and Web 3.0 other than blockchain that could support the FSC. These tech-nologies include Artificial Intelligence, Big Data analytics, RFIDs, NFC, (Internet of things) IoTs, Edge computing, Cloud Computing, and more that contribute to the overall growth in technologies that support the FSC.

Although Gartner (2022) predicted Top 10 Strategic Technology Trends for 2023 con-sisting of most of the upcoming industry 4.0 and web 3.0 technologies,  most of the recent blockchain-based food traceability and supply chain frameworks ( Hidayati et al., 2023; Akazue, Gupta et al., 2023; Babu et al., 2023; Ekawati et al., 2023; Tao et al., 2023; Jadav et al., 2023;  Taloba et al., 2023) incorporated only a few interoperable technologies with blockchain. However, upon reviewing the existing literature, it has been observed that the implementation of many of these emerging technologies, such as Artificial intelligence, Data analytics, Edge computing, Digital Twins NTFs and Metaverse, is scarce or absent in the current frameworks for blockchain-based food supply chains.

We are of the opinion that our research would make a significant contribution by en-abling researchers to delve into the interoperability of blockchain technology with emerg-ing Industry 4.0 and Web 3.0 technologies. This, in turn, would empower them to develop more advanced frameworks for blockchain-based food supply chains.”

Additionally, it would be beneficial to include a paragraph, at the end, that provides an overview of the content in the subsequent sections. This will guide the readers on what to expect and how the paper is structured.

Response:

As suggested, the following information has been added at the end of the introduction:

“The subsequent sections of this research paper are as follows, discussion about Emerging technologies with respect to food supply chain and traceability followed by methodology, findings and discussion, limitations, conclusion and Annexures.”

Section 2 lacks substantial and insightful content.

Response:

Thank you. Section 2 that is methodology has now been named as Section 3, we have added the following information to make the methodology more understandable and insightful. Our approach is the same that has been systemically used by other reviewers and PRISMA as stated in Lines # 333-340 “We followed the guidelines outlined by Kitchenham and Charters (2007) and Tranfield et al. (2007) and adhered to the recommendations of Moher et al. (2009) regarding the "Preferred Reporting Items for Systematic Reviews (PRISMA)." Systematic reviews involve the use of explicit and reproducible methods to address predefined research questions by identifying, critically evaluating, and combining findings from primary research studies.

To address the comments, we have added the following information:

“This systematic review adheres to a comprehensive approach in order to fulfill its objec-tives. To begin with, it formulates a well-defined research question that serves as a guiding principle throughout the review process. Subsequently, criteria are established to deter-mine which studies should be included or excluded, ensuring that only pertinent ones are considered. The selection of literature involves a meticulous process of carefully choosing relevant studies and assessing their quality to guarantee reliability and validity. Once the literature is gathered, it undergoes thorough analysis and synthesis to uncover patterns, trends, and significant findings. Ultimately, the findings are disseminated through di-verse channels, such as research articles, reports, or presentations, to effectively reach the target audience.

This systematic review process guarantees a rigorous and methodical approach to ac-quire, evaluate, and present the existing evidence related to the application of interopera-ble technologies with blockchain for food traceability and supply chain.”

Section 3 lacks clarity and precision, containing excessive and irrelevant context. The authors seem more focused on expanding the paper rather than conducting a thorough analysis based on their investigation.

Response:

We value the suggestion, and the MS has been amended accordingly. Section 4 (Previously Numbered as section 3) has been updated as follows;

The following contents has been removed from “Findings and Discussion” to a new section “2. Emerging technologies with respect to food supply chain and traceability:” because we believe that information is required for the general readership to be able to follow the subsequent discussion relevant to these technologies. Therefore, the information has been summarized and has been shifted to section 2 and thus the word count of findings section has substantially been reduced and findings seem to be more relevant:

Following information has been reduced from findings section:

Before proceeding to subsequent sections of this paper, it is crucial for readers to grasp the concepts of Industry 4.0 and Web 3.0. This understanding is particularly significant as the systematic review of blockchain-based food traceability and supply chain frameworks has unveiled the incorporation of diverse Industry 4.0 and Web 3.0 technologies in conjunction with blockchain. These findings will be elaborated upon in the subsequent section of the paper dedicated to the discussion of the results.

The first three industrial revolutions were characterized using steam engines, electronics, and information technology, respectively. These revolutions led to the mechanization, mass manufacturing, and digitization of production (Rymarczyk et al., 2020). Industry 4.0, also known as IR4.0, is a new era of manufacturing that is characterized by the convergence of the physical and digital worlds. The combination of physical and digital systems is referred to as Cyber Physical Systems (CPS). These CPS would allow for interaction between digital infrastructure and tangible objects, which makes information and services simultaneously accessible for a variety of purposes (Jazdi et al., 2014). Industry 4.0 technologies include, but are not limited to, additive manufacturing (3D Printing), artificial intelligence (AI) , Global Positioning System (GPS), big data and analytics (BDA), RFIDs blockchain (BC), cloud computing (CC), internet of things (IoT), simulation, Augmented reality, Robotics, Unmanned vehicles, and Drones (Dalenogare et al., 2018; Bai et al., 2020)

This convergence is enabled by multiple technologies including a group of internet-based technologies such as artificial intelligence (AI), machine learning (ML), big data, decentralised ledger technology (DLT), meta verse, decentralised applications (dApps), Semantic Web, edge computing, NFTs etc. are collectively known as Web 3.0. It is essential to observe that Web 3.0 is a concept that is still evolving, and that its full realisation has not yet occurred. Diverse initiatives and projects are actively constructing the infrastructure and tools necessary for the decentralised web (Wan et al., 2023).

Several websites and software programs with the capacity to process information intelligently while exhibiting intellect like that of an individual exactly define it. Web 1.0 was characterised by inert websites and simple HTML pages, whereas Web 2.0 introduced interactive and dynamic content, social media platforms, and user-generated content. Web 3.0 aims to boost the security and privacy of the decentralised Internet through the application of blockchain technology. In this way, Web 3.0 seeks to address several Web 2.0's limitations and concerns, especially in terms of data proprietorship, privacy, and centralised control (Wan et al., 2023).

In essence, Web 3.0 is considered the next stage of internet evolution, using decentralised technologies like peer-to-peer networking and the blockchain. Compared to the present Web 2.0, which is currently regulated by a limited number of large technology companies, it ensures a more user-centric, secure, and transparent experience (Lin et al., 2023). Web 3.0 applications base themselves on decentralised networks encompassing several peer-to-peer nodes (servers), blockchains, or an integration of the two. These programs are referred to as decentralised apps (DApps), and the Web 3.0 community employs this term often. Web 3.0's removal of regional boundaries and centralised decision-makers from the Internet has the potential to democratise financial institutions. Because Web 3.0 apps are based on decentralised protocols, they are not under the jurisdiction of a single organisation. Applications become resilient to surveillance and exploitation as a consequence data is not kept on centralised systems, giving users greater control over it (Lin et al., 2023; Suryavanshi et al., 2023).

By enabling computers to analyse data in a similar manner to humans, artificial intelligence (AI) and machine learning (ML) allow the intelligent generation and dissemination of useful information tailored to the unique requirements of a user (Lin et al., 2023; Suryavanshi et al., 2023) without the interference of the big tech giants or given them access to your data or allowing them to decide what online information can be accessed. Web 3.0's widespread acceptance might herald a paradigm change in the way that people use the Internet today, opening up new avenues for creativity, teamwork, and financial empowerment.

Some of the most common technologies enabling Industry 4.0 and Web 3.0 are as follows:

  • Near-field communication (NFC), an established technology for over a decade, facilitates easy two-way transactions between electronic devices over short distances (Nguyen et al., 2019). The rapid expansion of the Internet of Things (IoT) has contributed to the widespread adoption of NFC. NFC consist of a reader and a tag, which activates upon entering the magnetic field generated by the reader's antenna coil. The tag then is activated and transmits data back to the receiver using NFC data exchange format (NDEF) message, harnessing energy from its own antenna (Rahimi et al., 2017). NFC is extensively utilised in applications such as contactless payment, transit cards, and door entry, as well as a broad range of other applications where basic data such as an identification number or text is swiftly and securely sent between two devices without the requirement for pairing.

  • Big data analytics is the extraction of insightful patterns and insights from complex and extensive collections of various data sets. It comprises using cutting-edge approaches like machine learning and data mining to find hidden patterns, allowing organizations to make wise decisions and gain an edge in many sectors (Misra et al., 2020; Krzyzanowski, 2019).

  • AI refers to a technological tool that imitates human intelligence and cognitive processes using machines, predominantly computer systems, robotics, and digital devices (Patel et al., 2021).

  • The Global Positioning System (GPS) is a global navigation system that employs a constellation of satellites to deliver highly accurate positioning and timing data to Earth's users. GPS empowers individuals and devices to ascertain their precise coordinates, velocity, and altitude with exceptional precision. It has emerged as a vital technology across numerous domains, such as vehicle navigation, tracking systems, mapping services, and outdoor pursuits (Antonucci et al., 2019; Yakubu et al., 2022).

  • Cloud computing encompasses a diverse range of computer services, such as virtualized infrastructure, storage, networking, and software applications, which are made available and managed remotely by a third-party provider over the internet. This paradigm enables users to scale their computing resources dynamically, pay for actual usage, and leverage the provider's infrastructure. As a result, customers benefit from increased flexibility, scalability, cost-effectiveness, and accessibility compared to traditional on-premises computing (Madhavaram & Bashir, 2012).

  • QR codes have become widely utilized in various domains such as product tracing, mobile payments, advertising, creating degrees and transcripts, passport verification, and more. These codes serve as effective means of transmitting information. In case of defacement, QR codes offer four user-selectable error correction levels (ECL) - L, M, Q, and H - capable of rectifying errors up to 7%, 15%, 25%, and 30% respectively. The QR code standard encompasses 40 symbol variations to accommodate diverse data payloads (Dey et al., 2022; Huang et al., 2020).

  • RFID technology is widely utilized by researchers investigating the traceability of the food supply chain. RFID is a flow control technology that enables the traceability of goods through all stages of the production chain using a few simple and inexpensive components. For example, a tag can be directly applied to the displacing good; an interrogator or antenna, as a device (the reader) that gathers information from the tag (ID or data stored); and a database system used to store the information gained through the interrogation routines carried out by the interrogator or antenna (Costa et al., 2014).

  • In recent years, the Internet of Things (IoT) has emerged as a key technology widely employed in the food supply chain's information administration. IoT is supported by embedded devices such as sensors, actuators, and network connectivity. (Hasan et al., 2023) These devices facilitate real-time communication and data exchange between various smart devices in the supply chain, thereby fostering seamless information sharing (Hasan et al., 2023).

  • According to Lopez et al. (2015) and Shi et al. (2016), edge computing is a revolutionary approach to computing that enables the implementation of computational tasks at the periphery of a network. In contrast to traditional cloud computing, in which cloud services manage downstream data and IoT services manage upstream data, Shi et al. (2016) introduced the concept of "edge," which includes computing and network resources located between data sources and cloud data centres. The underlying principle of edge computing is to execute computations near to the data sources. This computing paradigm has several advantages over cloud computing. As highlighted by Yi et al. (2015), it significantly reduces response time by relocating computations from the cloud to the periphery. As demonstrated by Chun et al. (2011) and Pan and McElhannon (2018), it also obtains significant energy efficiency gains of between 30 and 40 percent.

  • IPFS implements the Interplanetary Linked Data (IPLD) standards, facilitating the development of decentralised, globally addressable, adaptable, and flexible data structures for various data types. The Interplanetary Way back also functions as a persistent Web archive by disseminating data files across the IPFS network (Alam et al., 2016).

  • The Metaverse encompasses a network of interconnected virtual realms that coexist alongside and enhance the physical world. These virtual realms enable users, who are represented by avatars, to connect, interact, and engage with one another while immersing themselves in a persistent, synchronous environment enriched by user-generated content. To encourage contributions to the Metaverse, an economic framework is established, providing incentives for active participation (Weinberger, 2012).

I would suggest reworking this section by providing a concise and focused analysis. Also, presenting the findings in a structured format would greatly enhance the readers' understanding of the research output. 

Responses

Additionally, to make findings more focused and to depict a meaningful analysis Figure 6. Interoperable Web 3.0 and Industry 4.0 Technologies used in blockchain-based food traceability and supply chain frameworks has been added.

Furthermore, based on the suggestions of Reviewer 2, the following tables have been added;

In order to provide better outlook to the readers the below table gives insight of the main applications along with the advantages and disadvantages of major Industry 4.0 and Web 3.0 technologies: Please bear in mind that the level of security, scalability, and other parameters can vary depending on the specific implementations and configurations of these technologies.

Table 2. Overview of applications, advantages and disadvantages of major Industry 4.0 and Web 3.0 technologies:

Technology

Description

Main Applications

Advantages

Disadvantages

Blockchain

A decentralized digital ledger for secure transactions and data storage.

Cryptocurrency, supply chain management, smart contracts, finance

Transparency, security, immutability.

Scalability, energy consumption, storage requirements, regulatory concerns.

IoT

An interconnected network of physical devices and sensors that communicate and exchange data.

Smart homes, industrial automation, healthcare monitoring.

Automation, real-time insights, efficiency.

Security vulnerabilities, data privacy, interoperability.

RFID

Uses radio waves to identify and track objects with tags containing electronic information.

Inventory management, asset tracking, and access control.

Efficiency, real-time tracking, and reduced manual effort.

Cost, limited range, potential interference.

QR Code

Two-dimensional barcode that can store various types of data.

Marketing campaigns, ticketing, and payment systems.

Versatility, easy scanning, and high storage capacity.

Limited data capacity and scanning limitations in certain conditions.

Cloud Computing

Delivery of on-demand computing resources over the internet.

Infrastructure as a Service (IaaS), Platform as a Service (PaaS), and Software as a Service (SaaS).

Scalability, cost-effectiveness, accessibility.

Data security, vendor lock-in, and potential downtime.

Artificial Intelligence

Simulation of human intelligence in machines for autonomous learning and decision-making.

Natural language processing, computer vision, recommendation systems.

Automation, improved efficiency, advanced analytics.

Ethical concerns, bias, job displacement.

Big Data

Large and complex data sets that require specialized processing techniques.

Business analytics, predictive modelling, personalized recommendations.

Insights generation, competitive advantage.

Data privacy, storage infrastructure, data quality.

GPS

Global navigation satellite system for location and timing information.

Navigation, logistics, and geolocation services.

Accuracy, real-time tracking, and widespread availability.

Signal limitations indoors or in remote areas.

NFC

Short-range wireless communication for contactless data exchange.

Mobile payments, access control, and ticketing systems.

Convenience, simplicity, and compatibility with smartphones.

Limited range, security concerns, adoption barriers.

The below table depicts the chracterstics, evaluation and comparison table for major Industry 4.0 and Web 3.0

Table 3. Characteristics of major Industry 4.0 and Web 3.0 technologies:

Technology

Security features

Connectivity

Scalability

Data handling

Blockchain

Offers high-level security with decentralized networks

Enables connectivity among multiple devices

Scales well for transactions and data storage

Handles data immutability and transparency through a ledger

IoT

Exhibits varying levels of security

Facilitates connectivity across diverse devices and protocols

Scales effectively for large-scale deployments

Manages real-time data collection and analysis

RFID

Provides basic security measures

Utilizes short-range wireless communication

Supports scalability for tracking multiple items

Offers limited data storage capacity

QR Code

Lacks inherent security features

Enables connectivity through scanning

Not applicable for scalability

Encodes and retrieves data efficiently

Cloud Computing

Implements robust security measures

Utilizes internet-based connectivity

Highly scalable for resource allocation

Manages remote data storage and processing

Artificial Intelligence

Security implementation varies

Relies on internet-based connectivity

Scales effectively for data processing and analysis

Handles large-scale data processing and learning algorithms

Big Data

Security implementation varies

Relies on internet-based connectivity

Highly scalable for handling large datasets

Manages data storage, processing, and analysis

GPS

Lacks inherent security features

Utilizes global positioning and satellite connectivity

Not applicable for scalability

Tracks location-based data effectively

Such a broad conclusion is not necessary. The conclusion should solely focus on the research output. Also, please use the appropriate tense in the conclusion. Furthermore, the future scope of the research is not very clear.

Response:

Thank you for the comment. The size of the conclusion has been reduced and the following text has been added to depict the contribution and future scope of the research:

“At the moment most of the blockchain-based food traceability and supply chain frame-works including the recent one fail to implement the interoperability of blockchain with emerging Industry 4.0 and Web 3.0 technologies. The reason for such a gap in the applica-tion of technology is due to web 3.0 and Industry 4.0 technologies are still in development and the application of these technologies is yet to be incorporated with respect to the food supply chain.

Our research provides a valuable contribution by opening new avenues of research to explore the interoperability of blockchain technology with emerging Industry 4.0 and Web 3.0 technologies and will enable researchers to delve into the interoperability of blockchain technology with emerging technologies. This, in turn, would empower future researchers to develop more advanced frameworks for blockchain-based food supply chains”

Reviewer 2 Report

Blockchain-based frameworks for food traceability is an interesting topic worth studying. However, as far as the quality of this paper is concerned, the quality of this paper is general. Here are some comments or suggestions for improving the current version.

1. Using the excellent characteristics of blockchain technology to solve the problems existing in traditional traceability has great advantages, but the blockchain is also facing problems such as insufficient storage space and excessive storage pressure. A systematic review of this topic should be more comprehensive, but this article lacks an objective evaluation of blockchain issues.

2. In fact, food traceability data has the characteristics of multi-source heterogeneity, complexity and diversity, and it seems that no one technology can be applied universally. Therefore, the objective evaluation of the advantages and disadvantages of each technology, and the comparison of various technical views, will make readers more able to correctly adopt the value of one or several technologies.

3. The method of literature retrieval is the cornerstone of obtaining literature information accurately. At present, there are some omissions in the search command in this paper, such as the command about food, such as vegetables, grains, aquatic products, milk, bee products, catering and other aspects of the search literature. It is suggested to combine several literature retrieval methods, such as citespace, to ensure more comprehensive literature data acquisition.

Author Response

Dear the Editor and Reviewer 2,

Thank you very much for your time and effort in reviewing our paper that improved our MS. We carefully addressed the comments and responded to all queries.

All new and amended text are highlighted in yellow for ease of tracking.

Shifted and reduced text is highlighted as blue in the manuscript.

Deleted text is stricken and is coloured red.

Best regards,

The Authors

Reviewer 2:

Using the excellent characteristics of blockchain technology to solve the problems existing in traditional traceability has great advantages, but the blockchain is also facing problems such as insufficient storage space and excessive storage pressure. A systematic review of this topic should be more comprehensive, but this article lacks an objective evaluation of blockchain issues.

Response:

We really appreciate this valuable suggestion. The following table has been added to give an overview about the disadvantages of Blockchain technology:

Table 2. Overview of applications, advantages & disadvantages of major Industry 4.0 and Web 3.0 technologies

Technology

Description

Main Applications

Advantages

Disadvantages

Blockchain

A decentralized digital ledger for secure transactions and data storage.

Cryptocurrency, supply chain management, smart contracts.

Transparency, security, immutability.

Scalability, energy consumption, storage Requirements, regulatory concerns.

IoT

Interconnected network of physical devices and sensors that communicate and exchange data.

Smart homes, industrial automation, healthcare monitoring.

Automation, real-time insights, efficiency.

Security vulnerabilities, data privacy, interoperability.

RFID

Uses radio waves to identify and track objects with tags containing electronic information.

Inventory management, asset tracking, access control.

Efficiency, real-time tracking, reduced manual effort.

Cost, limited range, potential interference.

QR Code

Two-dimensional barcode that can store various types of data.

Marketing campaigns, ticketing, payment systems.

Versatility, easy scanning, high storage capacity.

Limited data capacity, scanning limitations in certain conditions.

Cloud Computing

Delivery of on-demand computing resources over the internet.

Infrastructure as a Service (IaaS), Platform as a Service (PaaS), Software as a Service (SaaS).

Scalability, cost-effectiveness, accessibility.

Data security, vendor lock-in, potential downtime.

Artificial Intelligence

Simulation of human intelligence in machines for autonomous learning and decision-making.

Natural language processing, computer vision, recommendation systems.

Automation, improved efficiency, advanced analytics.

Ethical concerns, bias, job displacement.

Big Data

Large and complex data sets that require specialized processing techniques.

Business analytics, predictive modelling, personalized recommendations.

Insights generation, competitive advantage.

Data privacy, storage infrastructure, data quality.

GPS

Global navigation satellite system for location and timing information.

Navigation, logistics, geolocation services.

Accuracy, real-time tracking, widespread availability.

Signal limitations indoors or in remote areas.

NFC

Short-range wireless communication for contactless data exchange.

Mobile payments, access control, ticketing systems.

Convenience, simplicity, compatibility with smartphones.

Limited range, security concerns, adoption barriers.

In fact, food traceability data has the characteristics of multi-source heterogeneity, complexity and diversity, and it seems that no one technology can be applied universally. Therefore, the objective evaluation of the advantages and disadvantages of each technology, and the comparison of various technical views, will make readers more able to correctly adopt the value of one or several technologies.

Response:

Thank you for your valuable comment. We totally agree with the above point and in order to address this concern, the following table has been added:

The below table depicts the characteristics, evaluation and comparison table for major Industry 4.0 and Web 3.0

Table 3. Characteristics of major Industry 4.0 and Web 3.0 technologies:

Technology

Security features

Connectivity

Scalability

Data handling

Blockchain

Offers high-level security with decentralized networks

Enables connectivity among multiple devices

Scales well for transactions and data storage

Handles data immutability and transparency through a ledger

IoT

Exhibits varying levels of security

Facilitates connectivity across diverse devices and protocols

Scales effectively for large-scale deployments

Manages real-time data collection and analysis

RFID

Provides basic security measures

Utilizes short-range wireless communication

Supports scalability for tracking multiple items

Offers limited data storage capacity

QR Code

Lacks inherent security features

Enables connectivity through scanning

Not applicable for scalability

Encodes and retrieves data efficiently

Cloud Computing

Implements robust security measures

Utilizes internet-based connectivity

Highly scalable for resource allocation

Manages remote data storage and processing

Artificial Intelligence

Security implementation varies

Relies on internet-based connectivity

Scales effectively for data processing and analysis

Handles large-scale data processing and learning algorithms

Big Data

Security implementation varies

Relies on internet-based connectivity

Highly scalable for handling large datasets

Manages data storage, processing, and analysis

GPS

Lacks inherent security features

Utilizes global positioning and satellite connectivity

Not applicable for scalability

Tracks location-based data effectively

The method of literature retrieval is the cornerstone of obtaining literature information accurately. At present, there are some omissions in the search command in this paper, such as the command about food, such as vegetables, grains, aquatic products, milk, bee products, catering, and other aspects of the search literature. It is suggested to combine several literature retrieval methods, such as cite space, to ensure more comprehensive literature data acquisition.

Response:

Based on your comments in order to ensure maximum capture of data search string has been updated as:

(TITLE-ABS-KEY ("Blockchain” OR “Block chain"  OR  "distributed ledger")  AND  TITLE-ABS-KEY ("Food"  OR  "Meat"  OR  "Agri"  OR "Dairy" OR "Vegetables" OR "Fruits"     OR  "Grains"  OR  "Fish" OR "Honey"  OR  "Milk"  OR  "Fish"  "Tea"  OR  "Coffee"  OR  "drink"))

The change in search string resulted in 321 additional records and thus initial results that were mentioned in Prisma diagram were increased from 1231 records to 1552 records, however after removing duplicates and after screening 54 studies have been included in review as compared to 51 before these additional 3 studies have been updated in the Annexure as well

Round 2

Reviewer 1 Report

Thank you for considering my comments. The paper has indeed been improved, but there are still many areas that need further attention.

First and foremost, rather than elongating the paper, the authors should focus on enhancing the quality. There are too many unnecessary contexts that overshadow the main contribution. 

Secondly, I recommend avoiding providing  overly broad definitions of other authors' work/methods. A concise 1-2 line small discussion/review should suffice. If readers need further information about those methods, they can always refer to the cited sources to obtain further information. The main focus of the research should be the analysis derived from the review. 

Understandably, this is a review paper, but the way the paper is structured may confuse readers to obtain valuable information. Thus, the paper requires significant improvement. 

I would suggest reviewing more  survey papers and adopting their structure as a guide. Additionally, the authors can also review the following paper for insights on the structure; 

Assessing Blockchain Consensus and Security Mechanism Against the 51% Attack. Sarwar Sayeed, Hector Marco-Gisbert

Author Response

Manuscript ID: foods-2488703 Response letter

Dear the Editor and Reviewer 1,

Thank you very much for your time and effort in reviewing our paper that improved our MS. We carefully addressed the comments and responded to all queries.

Best regards,

The Authors

Reviewer 1

Thank you for considering my comments. The paper has indeed been improved, but there are still many areas that need further attention.

First and foremost, rather than elongating the paper, the authors should focus on enhancing the quality. There are too many unnecessary contexts that overshadow the main contribution.

Secondly, I recommend avoiding providing overly broad definitions of other authors' work/methods. A concise 1-2 line small discussion/review should suffice. If readers need further information about those methods, they can always refer to the cited sources to obtain further information. The main focus of the research should be the analysis derived from the review.

Understandably, this is a review paper, but the way the paper is structured may confuse readers to obtain valuable information. Thus, the paper requires significant improvement.

I would suggest reviewing more survey papers and adopting their structure as a guide. Additionally, the authors can also review the following paper for insights on the structure.

Response:

Thank you for your valuable feedback and suggestions regarding our manuscript.

Your suggestion of editing the content judiciously to increase the visibility of the primary scientific points is astute. We understand that conciseness and clarity are crucial in effectively communicating our research. By carefully refining the content, we have aimed to retain the value of the manuscript while presenting the information in a more accessible manner.

We have worked diligently to strike the right balance between maintaining the value and shortening the length of the manuscript from 22000 words to around 15000 words. In the revised MS, we aimed at achieving the appropriate combination or ratio for better readability enable readers to grasp the key insights without being overwhelmed by unnecessary details.

Once again, we express our gratitude for your thoughtful review and constructive comments.

Reviewer 2 Report

I think the authors have resolved my comments.

Author Response

Thank you very much for your time and effort in reviewing our paper that improved our MS.